# ViSioNS: Visual Search in Natural Scenes Benchmark

**Fermin Travi**[1,2*]
ftravi@dc.uba.ar

**Gonzalo Ruarte**[1*]
gruarte@dc.uba.ar

**Gaston Bujia**[1,2]
gbujia@dc.uba.ar

**Juan E. Kamienkowski**[1,2,3]
juank@dc.uba.ar

[1] Laboratorio de Inteligencia Artificial Aplicada, Instituto de Ciencias de la Computación, Universidad de Buenos Aires - CONICET, Argentina; [2] Depto. de Computación, Fac. de Cs. Exactas y Naturales, Universidad de Buenos Aires, Argentina; [3] Maestría de Explotación de Datos y Descubrimiento del Conocimiento, Universidad de Buenos Aires, Argentina
[*]Both authors contributed equally to the present work.

## Abstract

Visual search is an essential part of almost any everyday human interaction with the visual environment [1, 2]. Nowadays, several algorithms are able to predict gaze positions during simple observation, but few models attempt to simulate human behavior during visual search in natural scenes. Furthermore, these models vary widely in their design and exhibit differences in the datasets and metrics with which they were evaluated. Thus, there is a need for a reference point, on which each model can be tested and from where potential improvements can be derived. In this study, we select publicly available state-of-the-art visual search models and datasets in natural scenes, and provide a common framework for their evaluation. To this end, we apply a unified format and criteria, bridging the gaps between them, and we estimate the models' efficiency and similarity with humans using a specific set of metrics. This integration has allowed us to enhance the Ideal Bayesian Searcher by combining it with a neural network-based visual search model, which enables it to generalize to other datasets. The present work sheds light on the limitations of current models and how integrating different approaches with a unified criteria can lead to better algorithms. Moreover, it moves forward on bringing forth a solution for the urgent need of benchmarking data and metrics to support the development of more general human visual search computational models. All of the code used here, including metrics, plots, and visual search models, alongside the preprocessed datasets, are available at `https://github.com/FerminT/VisualSearchBenchmark`.

## 1   Introduction

The neural mechanisms underlying the visual processing of an image at the center of the visual field, without eye movements, have been thoroughly studied [3, 4]. This knowledge inspires the development of precise models of image processing and feature extraction (both objective and subjective) in the field of Artificial Intelligence [5, 6, 7]. Closing the loop, these models also provide an intuition to the understanding of the algorithms that underpin human cognition [7, 8, 9, 10]. Among their recent successes, they have been highly effective at predicting gaze positions over an image when it is being freely explored [11, 12, 13, 14]; the metric-specific predicted spatial distribution, derived from the model's fixation density map, is called a saliency map [15, 16].

36th Conference on Neural Information Processing Systems (NeurIPS 2022) Track on Datasets and Benchmarks.

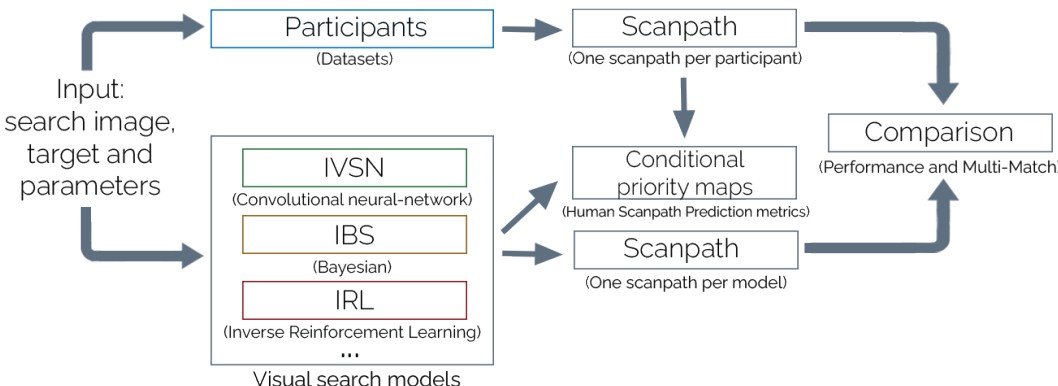

Figure 1: Methodology of the present work. IVSN, IBS, and IRL refer to different visual search models (Invariant Visual Search Network, Ideal Bayesian Searcher, and Inverse Reinforcement Learning respectively). Parameters could include, for instance, the target category (instead of a template) for IRL.

Nonetheless, there are two fundamental aspects where current computational models are yet to provide answers. First, when a biological system (mainly humans) observes an image, it is generally done with a goal in mind or to fulfill a task, such as memorizing the image or searching for an object (or a specific category of objects). Second, it is not only relevant where the eye fixates upon, but also in what order it scans the image: eye movements do not follow random behavior, but rather different strategies that intend to minimize the number of fixations necessary to reach a goal [17, 18, 19, 20].

Currently, there are diverse algorithms that attempt to predict the sequence of eye movements (called *scanpath*) when looking for an object in a natural scene [21, 22, 23, 24]. However, the theoretical frameworks behind these models vary widely from each other. For instance, there are models that employ convolutional or recurrent neural networks [22, 23], others that utilize reinforcement learning [24], and a third category that applies a Bayesian approach [21, 25, 26]. Due to the lack of a reference point, these models were tested exclusively on their own datasets. Moreover, despite these datasets are used for making similar claims on the visual search task in natural scenes, they can be categorized by the difficulty of the task (measured by the number of fixations made to find the target), the content of the images (absence or presence of text, human faces, and other distractors), the task itself (looking for a specific target or any element from a given category), how the target is presented, and so on. Further variation can also be found in the methodologies employed to evaluate the results obtained. Altogether, these differences hinder the task of concluding what works best and what needs further study. In the past decade, saliency models for the visual exploration task greatly benefited from the construction of a saliency benchmark [27, 16],[1] and the field of visual search is in need of one.

In the present work, we aimed to compare state-of-the-art visual search models in natural scenes. We performed a systematic search and selected the three models that were publicly available and allowed natural scenes images as input [21, 22, 24], alongside the datasets they were tested on [28], with the addition of another published dataset [23]. Notwithstanding, we intend to facilitate the incorporation of new models and data by releasing all of the code and preprocessed datasets.[2] Additionally, in order to achieve a correct comparison, we defined a common set of metrics and baseline models [28, 29, 30, 31]. In sum, we bring together different approaches to the problem of visual search in natural scenes in a common computational framework and evaluation criteria (Fig. 1). By doing so, we were able to improve a state-of-the-art visual search model by integrating Bayesian inference with neural networks. Finally, we create a reference point that provides further insight on how to move forward.

---

[1]https://saliency.tuebingen.ai/
[2]https://github.com/FerminT/VisualSearchBenchmark

## 2 Methods

### 2.1 Datasets

The selected datasets are, to our knowledge, the only ones on the task of visual search in natural scenes that are publicly available. Each dataset comprises a different set of curated search images, target objects for each image, and anonymized participants' scanpaths on those images. They can be categorized mainly based on the following characteristics (Table 1).

Table 1: Main characteristics of the datasets considered, highlighting the differences between them. Most datasets contain distractors of some sort, but, in this case, we are focusing on objects belonging to the same category as the target. Only scanpaths where the target was found are taken into account. Mean scanpath length is expressed in number of fixations with the interquartile range in parentheses.

|                        | Interiors  | Unrestricted   | MCS          | COCOSearch18 |
|------------------------|------------|----------------|--------------|--------------|
| #Participants per image| 57         | 15             | 2-3          | 10           |
| #Images                | 134        | 234            | 1687         | 612          |
| #Scanpaths             | ∼3.7K      | ∼2.7K          | ∼4.2K        | ∼5.5K        |
| Mean scanpath length   | 5.2 (4, 6) | 10.7 (4, 13)   | 3.9 (3, 5)   | 2.7 (2, 3)   |
| People                 | ×          | ✓              | ×            | ×            |
| Exact Target           | ✓          | ×              | ×            | ×            |
| Distractors            | ✓          | ×              | ×            | ×            |
| Color                  | ×          | ×              | ✓            | ✓            |
| Fovea size             | 32x32      | 45x45          | 20x20        | 52x52        |
| Area covered by target | 0.6%       | 0.03% to 8.7%  | 0.02% to 38% | 0.9% to 9.8% |
| Image resolution       | 768x1024   | 1024x1280      | 508x564      | 1050x1680    |

**Target presentation:** The *Interiors* dataset (which corresponds to the IBS family models) uses exact templates as targets and observers have to look for a specific object [21, 31]. Both *MCS* and *COCOSearch18* state the object category ("microwave" and "clock" in the former, "car", "bottle" and others in the latter) and participants have to look for a broad object category (which could be in the image or not) [23, 28], and in the *Unrestricted* dataset (the one used for the IVSN model in natural scenes) a generalized version of the object is presented, so participants have to look for an object in a narrower category defined by the template (for example, a shoe or a stroller) [22].

**End of search criteria:** Not all of the humans' scanpaths finish just as soon as a fixation from the participant lands on the target. In *MCS*, *COCOSearch18*, and the *Unrestricted* dataset, participants have to press a button to finish the trial, so they can fixate on the target many times before they decide it is there. Conversely, in the *Interiors* dataset, the trial finishes when the participants fixate on the target or after a varying upper limit of saccades is reached.

**Target found criteria:** When compared to their corresponding models, the authors of *MCS* and *COCOSearch18* decide that the target was found if the average position during a fixation lands inside the target's bounding box (with no regard for the fixational eye movements or the size of the fovea), while Sclar et al. reduce the fixations to a grid using cells whose size is estimated from the fovea [21], and Zhang et al. use a square window, centered on the current target, whose size corresponds to two times the mean width and height of all the targets' bounding boxes [22].

**Image content:** The selection of images in the *Unrestricted* dataset was completely unconstrained and it is the only dataset to include human figures (faces or bodies) in them. Meanwhile, the *Interiors* dataset only includes images of crowded interiors which could contain several objects belonging to the same category as the target (distractors), and both *MCS* and *COCOSearch18* are the only ones with colored images.

**Dataset size:** There was a huge variability in the number of participants per image (2 to 57), images (134 to 1687), and the size of the targets (with the average area covered ranging from 0.6% to 4%).

In light of all these differences in experimental design, we carried out a preprocessing of these datasets in order to fit them into a common format and criteria:

- Targets are regarded as having been found if the area covered by a given fixation (determined by an approximation of the size of the fovea) lands on the bounding box of the target.

- Consecutive fixations whose distance between each other is less than the estimated size of the fovea are lumped together, to allow for a fairer comparison with the computational models (see Appendix A.2).

- Scanpaths are cut as soon as the target is found. For visual search models, this means the search ends whenever this happens or when an upper saccade limit is reached.

- Search images where the initial fixation lands on the target are excluded.

- Target templates, as well as their category and bounding boxes, are provided for all images.

For a more detailed list of the changes performed, please refer to Appendix A.1.

## 2.2 Models

### 2.2.1 Visual Search Models

We focused on visual search models that support natural (i.e., not synthetic) images as input (Fig 2). Among those, as was done with the datasets, we selected the ones that released their code. The first criteria excludes, for instance, [25] and [26], and the second one excludes [23], although they can still be added in the future by either a combination with the other models or when their code is released.

**Invariant Visual Search Network (IVSN) [22]**[3]**:** IVSN is based on convolutional neural networks inspired by the human visual cortex and performs a zero-shot invariant visual search. It makes use of two different versions of a pretrained VGG16: one for the search image, which is used to extract bottom-up features (visual ventral cortex network), and a second one for the target image, which is used to extract top-down features (prefrontal cortex network). Once these extractions have been done, the feature representation of the target image is used to modulate the representation of the search image through a convolution, generating what Zhang et. al. called an attention map [22]. A greedy search is then performed on this attention map, which means the whole image is processed at once (contrary to how human vision works), there is no accumulation of information across saccades, and the inhibition-of-return (avoiding fixating locations that have already been observed) [32] is forced.

**Ideal Bayesian Searcher (IBS) [21, 31]**[4]**:** The correlation-based IBS (cIBS) attempts to minimize the number of fixations needed to find the target. First, it uses a saliency map of the search image computed with DeepGaze II [33] as prior. Then, at each step, the next fixation is calculated as the one that maximizes the likelihood of finding the target, taking into account all the information gathered in previous steps [34]. The likelihood is calculated through a visibility map (estimated as a 2D Gaussian centered on the current fixation) and a target similarity map, which is computed with the search image and the target template via cross-correlation [21]. Given the opportunity to test this model in several different datasets, we evaluated different versions, modifying its target similarity map. In particular, we replaced cross-correlation with either Structural Similarity Index (SSIM) [35, 31] or a convolutional neural network (sIBS and nnIBS, respectively). The latter makes use of the attention map computed by IVSN and, by performing this substitution, the resulting model is able to capture object invariance while maintaining a good performance on template matching. This variation was used against the other models.

**Inverse Reinforcement Learning (IRL) [24]**[5]**:** IRL was trained on *COCOSearch18*, performing categorical visual search, and it is only capable of searching for an object that belongs to one of those 18 categories. Similar to cIBS, a Gaussian blur is applied to the search image to account for the loss of visibility in the periphery. Both, the unaltered search image and the blurred image, are preprocessed with a neural network pre-trained on COCO2017 which performs a panoptic segmentation (Detectron2 [36]) and its output, the so-called high and low resolution belief maps, are used as the model's input. On top of this, a one-hot encoding of the target object category is concatenated with the input of each layer in the network. This way, the authors aim to capture both the image and the target object's context. At the time of writing, the code for preprocessing the

---

[3]`https://github.com/kreimanlab/VisualSearchZeroShot`
[4]`https://github.com/gastonbujia/VisualSearch`
[5]`https://github.com/cvlab-stonybrook/ScanpathPrediction`

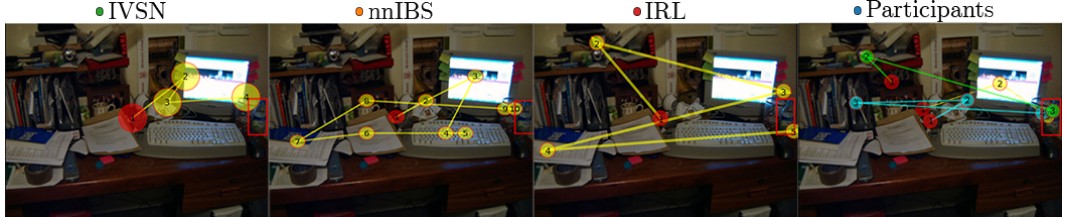

Figure 2: Examples of different scanpaths in the same search image, where the target is a bottle of water. Initial fixations are in red; scanpaths from different participants are superimposed with different colors. The size of each circle represents the area covered by the fixation.

images was not available, so we reproduced it from the references in the publication (see Appendix A.5 for details).

All of the models described were ported to Python (whenever necessary) in order to place them in a common framework. Additionally, the necessary changes to run them on the previously established dataset's format were made (Appendix A.2). Therefore, each model is now able to receive any of the four preprocessed datasets as input and the output follows the same criteria as that of participants', allowing for a correct comparison.

### 2.2.2 Baseline Models

We propose three different baseline models, similar to those in Kümmerer et. al. [30], with the goal of providing a lower and upper bound to the performance of the evaluated models. The lower bound is determined by the uniform and center bias models, and the upper bound by the Gold Standard model (GS). The uniform model predicts fixations to be uniformly and independently distributed over the image. The center bias model stems from Tatler et. al. [37], who show that people have a tendency to look at the center of images. As this phenomenon occurs mainly during free-viewing experiments [38], we use a Gaussian Kernel Density Estimate (GKDE) over all publicly available fixations in all images of the CAT2000 training dataset [39], excluding the first fixation as it is forced. Lastly, the Gold Standard model predicts fixations of each participant by means of a GKDE over all fixations of other participants on the same image [40]. In these last two cases, the bandwidth of the GKDE has been selected to yield maximum log-likelihood in a 5-fold cross validation paradigm.

### 2.3 Metrics

Prior work on visual search has focused on two main measures: the ratio of targets found accumulated across fixations (also referred as "cumulative performance" [22] or "cumulative target-fixation probability" [28]) and vectorized similarity between scanpaths by using Multi-Match [29, 31]. Even though these are useful for their specified purpose, they do not take into account the decision process that occurs fixation-by-fixation. To address this, we add human scanpath prediction (HSP), a recently published method for measuring the models' ability to predict participants' fixations that was previously employed in the visual exploration task [30] and is yet to be applied to visual search. By considering each fixation instead of the whole scanpath, this method also increases the dataset size by a factor of the mean scanpath length (see Table 1).

**Efficiency:** We use cumulative performance, measuring the proportion of targets found for a given number of fixations, and the area under this curve (AUC, here referred as AUC*perf* [28]) is reported.

**Scanpath similarity:** Multi-Match (MM) [29] is a pair-wise similarity metric that represents scanpaths as vectors in a two-dimensional space (any scanpath is built up of a vector sequence in which the vectors represent saccades, and the start and end position of saccade vectors represent fixations). Two such sequences (which can differ in length) are compared on four dimensions: vector shape, vector length (saccadic amplitude), vector position, and vector direction for a multidimensional similarity evaluation. The temporal dimension is excluded as we are not considering fixations' duration.

For each model, its scanpaths were compared to each participant's scanpaths (after reducing them to match the model's scanpaths dimensions) and the outcome was averaged across all dimensions and participants, resulting in a single scanpath similarity value for each image (human-model Multi-

Match or hmMM). The same was done with the participants' scanpaths, comparing them within themselves, and thus obtaining a human ground truth for each image (within-human Multi-Match or whMM) (Appendix Fig. A6). These operations were performed for every scanpath with length greater than two in which the target was found. We report AvgMM as the mean value of hmMM, in the case of models, and whMM, in the case of participants, over all images (Appendix Tables A2, A4). Considering that the difficulty of the search varies across images, we plotted whMM against hmMM to compare the variability throughout the dataset (Fig. 4D-E, 5E-H) and their correlation was calculated. These plots provide an easy overview of how well the model has captured human behavior: the closer the points are to the diagonal, the more exchangeable the model's scanpaths are with those of participants', whereas the points further to the left represent those images in which within human similarity was greatest, but human-model similarity was not as high [31].

**Human scanpath prediction (HSP):** For a given participant's scanpath, we evaluate how well each model predicts the next human fixation based on the scanpath history [30]. The key idea behind this method is to force the models to follow the participant's scanpath (ignoring its own predictions). At each step, each model creates what is called a "conditional priority map" (a priority map based on the participant's previous fixations from where the next fixation is sampled) and we compare the position where the model's fixation would land (i.e. its prediction of the next fixation) against the participant's fixation [30]. By using the latter as the ground truth, this allows for the computation of well-established metrics (such as AUC[41], NSS (Normalized Scanpath Saliency) [42], and Information Gain [40] relative to the center bias and uniform models; we refer to their average across fixations as AUC$hsp$, NSS$hsp$, IG$hsp$ and LL$hsp$, respectively. See Appendix A.7 for details). In this way, contrary to Multi-Match (which treats whole scanpaths as vectors), we are assessing the model's similarity to human behavior fixation-by-fixation, following the decision process (Fig. 3). To calculate these metrics, each model had to be adjusted to compute the conditional priority maps. IVSN is the simplest case, as there is no accumulation of information across saccades, so its WTA algorithm is made to select the participant's fixation instead of the maximum. For the IBS family models, when computing the likelihood, the participant's fixation is used in the visibility and target similarity maps. Lastly, for IRL, the participant's fixation is employed when updating the model's inner state, so both the inhibition-of-return effect and retrieval of the high resolution belief maps are done at that location. In these last two cases, as the search is carried out in grids of reduced dimensionality, it may happen that the target is found before the participant did in the original resolution. We opted for cutting short the participant's scanpaths whenever this occurred to minimize its impact on the metrics.

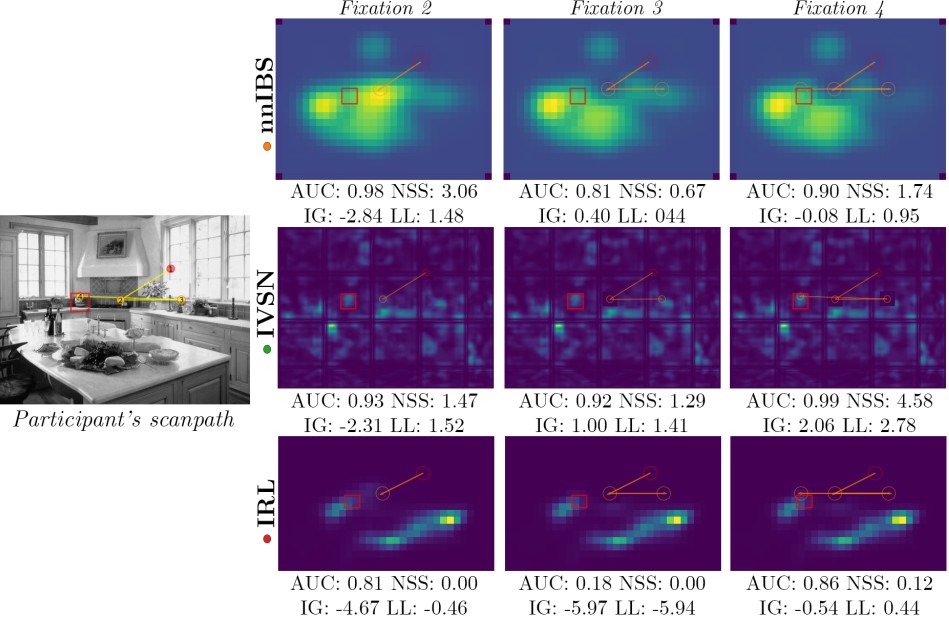

Figure 3: On the left, a given participant's scanpath. On the right, the conditional priority maps computed by each model at each step and the resulting AUC, NSS, IG and LL values.

Since the baseline models are not visual search models themselves (i.e. they don't generate scanpaths), only the fixation-by-fixation metrics (HSP) are computed on these. Due to the novelty of the evaluation method, this also serves the purpose of providing a reference frame. Finally, in order to sort the models by a compound variable, we defined their score as the average of the individual normalized metrics, relative to Humans in the case of efficiency and scanpath similarity metrics (AUC*perf*, AvgMM) or by the Gold Standard in the case of the scanpath prediction metrics (AUC*hsp*, NSS*hsp*, IG*hsp*, LL*hsp*). For AUC*perf*, this is done by performing $1 - |AUCperf_{Model} - AUCperf_{Humans}|$, as we intend to maximize similarity to human performance, and the rest of the scores are computed as $(Value_{Model} - Value_{Reference})/Value_{Reference}$, where *Reference* stands for Humans in AvgMM and for GS in all other cases.

## 3    Results

Different sets of images, targets, and tasks present different challenges to the models. These datasets all have crowded images, full of objects similar to the target (with varying degrees of resemblance), and with a strong context. Moreover, the task was slightly different (looking for an exact object or for a category) and, thus, the presentation of the target that guides the search was different. Considering this, we evaluate each model in all datasets reporting all the proposed metrics.

**Improving models by combining IBS with attention maps:** The cIBS model was previously evaluated in the *Interiors* dataset, which required looking for an exact object by cropping the target from the image and using it as template [21]. Briefly, this model consists of two parts: the prior that gives it a first gist of the image and an initial context, and the IBS rule to update the probabilities of finding the target, which is based on a similarity map that compares each image patch with the target template. In order to allow the model to both generalize any target example and also to get confused/attracted by distractors (as humans are), we evaluated three different similarity maps: the correlation and the SSIM between each image patch and the target template (cIBS and sIBS [31], respectively), and the attention map from the IVSN model (nnIBS). Template-matching performance when using a crop of the target was greatest with SSIM (Fig. 4A) (albeit at a greater computational cost), while nnIBS was the most capable of performing categorical visual search (Fig. 4C). Nonetheless, there were not significant differences in the scanpath similarity metrics (Fig. 4D-F; Appendix Table A3) and in both AUC*hsp* and NSS*hsp* (Table 2). Rather, what sets nnIBS apart from cIBS and sIBS consistently throughout all the datasets are IG*hsp* and LL*hsp* (Appendix Table A2), suggesting the convolutional neural-network is better able to provide human-like bottom-up processing of visual information about the image and target than structural methods such as SSIM and cross-correlation.

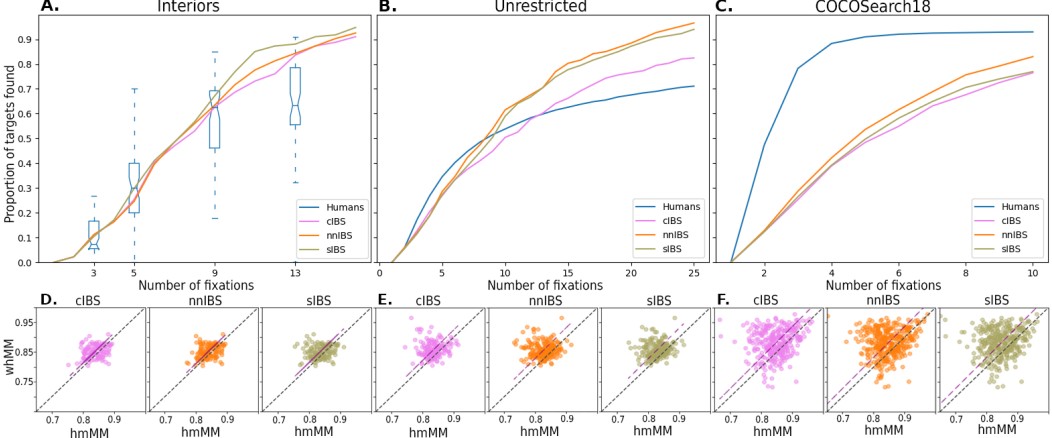

Figure 4: **A-C)** Ratio of targets found (vertical axis) for a given number of fixations (horizontal axis). In A), human performance is displayed as box plots due to the upper limits set on the number of saccades for participants (varying between 2, 4, 8 and 12). **D-F)** Mean MM values within humans (whMM) as a function of the mean MM values between each human observer and a model (hmMM) (Table A3). Each dot represents an image of the dataset considered. Only trials where the target was found are considered. Results for the *MCS* dataset can be found in Fig. A7.

Table 2: Average values across all datasets. Baseline models do not generate scanpaths, so cumulative performance (AUC*perf*) and Multi-Match (AvgMM and the resulting correlation, Corr) were not measured. *hsp* refers to human scanpath prediction, and AUC, NSS, IG, and LL correspond to the well-established saliency metrics (see Appendix A.7). The score is calculated as described in 2.3.

| | AUC*perf* | AvgMM | Corr | AUC*hsp* | NSS*hsp* | IG*hsp* | LL*hsp* | Score |
|---|---|---|---|---|---|---|---|---|
| • Humans | 0.56 | 0.87 | - | - | - | - | - | - |
| Gold Standard | - | - | - | 0.90 | 2.65 | 1.93 | 1.95 | 0.00 |
| • nnIBS | **0.55** | 0.84 | 0.15 | 0.74 | **1.27** | **0.44** | **0.35** | -0.17 |
| • cIBS | 0.51 | **0.85** | **0.17** | **0.75** | 1.26 | 0.31 | 0.23 | -0.19 |
| • sIBS | 0.54 | 0.84 | 0.13 | 0.74 | 1.25 | 0.31 | 0.23 | -0.19 |
| Center Bias | - | - | - | 0.72 | 0.89 | 0.00 | 0.07 | -0.70 |
| Uniform | - | - | - | 0.50 | 0.00 | -0.07 | 0.00 | -0.87 |
| • IVSN | 0.67 | 0.80 | 0.09 | 0.61 | 1.07 | -4.29 | -4.18 | -0.91 |
| • IRL | 0.40 | 0.80 | 0.04 | 0.65 | 1.24 | -4.83 | -4.90 | -1.00 |

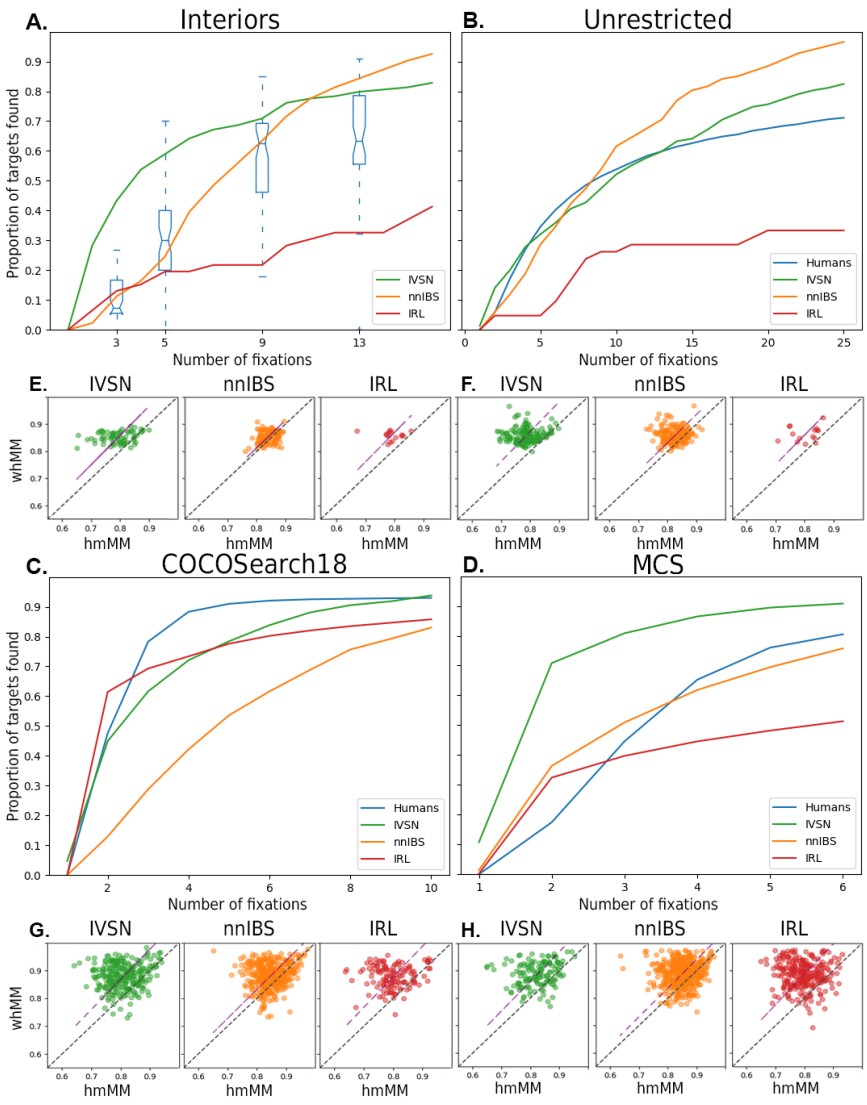

Figure 5: **A-D)** Cumulative performance curve for each model in each dataset. In A-B), IRL's performance is relative to the number of images that could be evaluated on. **E-H)** Mean MM values within humans (whMM) as function of the mean MM values between each human observer and a model (hmMM) (Appendix Table A5). Each dot represents an image of the dataset considered. Only trials where the target was found are considered.

**Visual search models:** IVSN was also modified by changing the patch size used for inhibition-of-return and target identification, to adapt it to the different datasets involved. This model performs a greedy search on an attention map built with neural networks. After an image patch is explored, the probabilities in that position are zeroed, forcing an inhibition-of-return effect [22]. Thus, the model is strongly dependent on this patch size. We evaluated different values for it relative to the mean target size of each dataset, including the one proposed by the authors, and kept the best working version (Appendix B.2).

When comparing all the visual search models, not surprisingly, each one performed best in its own dataset. As IRL depends on the categories of the *COCOSearch18* dataset, it could only be evaluated on a small portion of the *Unrestricted* and *Interiors* datasets. In both datasets, IRL's performance was particularly low, achieving under 50% accuracy (19 of 46 and 17 of 42 successful trials in the *Interiors* and *Unrestricted* datasets, respectively) (Fig. 5A-B). Additionally, even though it could run on the entire *MCS* dataset (for it only contains microwaves and clocks as targets), it didn't perform as well as the other models and participants (Fig. 5D; Appendix Table A4).

IVSN showed great search efficiency across all datasets, mainly in the first few fixations, likely as a consequence of its wider view of the scene –i.e. the model doesn't include a decay due to the foveated structure of the eye–. This, alongside its independence between fixations (there is no accumulation of information), is probably why it never achieved the highest score in neither the scanpath similarity (Appendix Table A5) nor the human scanpath prediction (HSP) metrics (Appendix Table A4).

The only visual search models to rank above the uniform and center bias models were those from the IBS family (Table 2), as they were the most consistent with human behavior in both scanpath similarity and scanpath prediction, with nnIBS standing in first place. Their whMM and hmMM values presented a significant correlation ($p < 0.05$) in all datasets but *Unrestricted*, something that doesn't hold true for IRL and IVSN (Appendix Table A4). Remarkably, even though IRL performed particularly well in *COCOSearch18*, it did not do so in IG*hsp* and LL*hsp*. There is a stark difference in these two metrics between the IBS family and IRL and IVSN across all datasets, implying the former is the only one capable of providing more information about human behavior during visual search than simple baseline models. However, it is still a far cry from the scores obtained by the Gold Standard, depicting how much work there is to be done in the realm of visual search in natural scenes models.

The uniform and center bias models are not agnostic to the datasets' design either. The latter performed particularly well in the *MCS* dataset, where there wasn't any criteria for selecting the target positions in the training set, resulting in many of them having targets close to the center. Moreover, the scoring difference between these two was significantly lower in the *Interiors* dataset, where initial fixations are randomized (Appendix Table A4).

**Further observations:** It is noteworthy how the distractors in the *Interiors* dataset seemed to be capable of fooling IVSN in some cases. That did not happen for nnIBS, even though it used the same attention map (Appendix Fig. A9). This could be due to its greedy behavior, which simply goes to the point that most closely resembles the target, regardless of distance or other factors (see also Appendix Fig. A10).

Additionally, in order to guide future improvements of visual search algorithms, we explored images where the models' scanpath dissimilarity with participants was greatest (the points farthest to the upper left in Fig. 5E-H). On one hand, participants were able to rapidly understand the context of the image they were looking at and performed their search accordingly [9, 43, 44]. For instance, while looking for a car, they immediately understood it would be on the road and not in the sky or buildings (Appendix Fig. A11A, see also Fig. A11B-C for other examples). Meanwhile, none of the visual search models analyzed here were able to capture this behavior.

On the other hand, it is interesting that participants did not follow salient distractors (such as human faces while looking for something else), whereas nnIBS did, mainly in the first fixations (Appendix Fig. A12). This is because of its prior, which is a saliency map computed with DeepGaze II [33], a neural network trained on participants' data in a free viewing task.

# 4 Conclusions

Even though all of the models and datasets evaluated here were designed for the task of visual search in natural scenes, the strong disparity in the models' performance among different datasets highlights how much work still needs to be done in terms of providing a clear definition of the problem at hand and how to assess the different solutions, as well as the importance of testing the visual search models in a diverse collection of datasets with comparable measures [45]. We offer the following key points for future experiment design and evaluation:

- **Task design:** Categorical visual search allows for an evaluation of the ability to capture object invariance, but fails to provide an account of the capacity to discern the target among several distractors belonging to the same category (as searching for a specific target does).
- **End of search criteria:** Finalizing the experiment as soon as a fixation lands on the target, instead of doing it by a button press, may fail to provide an account of when participants actively recognize it. Nevertheless, confidence measures could be added to confirm the response.
- **Target found criteria:** How a target is regarded as having been found needs to be clearly stated and properly justified. Here, we suggest using an estimation of the fovea size as the area covered by each fixation.
- **Image content:** Several objects and/or distractors should be included in the images and, ideally, these would be quantified. It is also important to note that, while colored images may be more realistic, they often result in shorter scanpaths than grayscale images due to the added information they possess (Table 1).
- **Number of participants:** Each trial should possess at least two or more participants. This is particularly important for the evaluation methods and for building a Gold Standard.
- **Target size and position:** Targets should never be placed near the center of the image, nor their size cover a large area of it, as it greatly reduces the difficulty of the task.
- **Model evaluation:** The most discernible differences stemmed from measuring the prediction of participants' fixations using well-established saliency metrics in conjunction with baseline models. Multi-Match, although consistent, did not provide such contrast.

Under these conditions, future datasets could be added to the benchmark and, after performing the preprocessing steps described above, different models could be fairly compared despite their variations in the visual search task.

With regard to the models evaluated, our work shows some benefits and drawbacks of each paradigm considered. On the one hand, the IRL approach seems to be limited at present, as it is able to incorporate context information, but it does not generalize outside of its own dataset. On the other hand, while the models based on DNNs considered here (such as IVSN) could achieve better search efficiency, the Bayesian approach produced the most human-like scanpaths and was the better predictor of participants' fixations. This is consistent with the idea of modeling human visual search as an active sampling process where successive steps attempt to reduce uncertainty about the localization of the target [46, 47]. Here, we suggest that combining these paradigms could potentially give rise to more precise and interpretable models. In particular, by modeling the visual system within a DNN framework [48], while applying a Bayesian approach for the central decision making [49, 50].

Notwithstanding, there is still plenty of room for improvement, as the scores of the Gold Standard depict. Particularly, combining scene content with the target's meaning remained elusive for these state-of-the-art visual search computational models and its incorporation could narrow the gap between them. Further work should be carried out in order to develop general visual search algorithms capable of capturing context. nnIBS could benefit from modifying its prior by creating a map that correlates the target with the image's context. Recently, Levi and Ullman [51] have proposed an efficient manner of using relational reasoning to build context for small object detection, aggregating information across the entire image. This approach, called Efficient Non Local module (ENL), might prove to be useful for this task. We are also interested in incorporating novel models to the pool, such as a recently released visual search model based on IVSN that includes eccentricity dependency [52].

The present work sheds light on the limitations of current models and how integrating different approaches with a unified criteria can yield potential improvements. Moreover, it is a first step toward solving the urgency for the definition of a common set of metrics and data for the development of more general human visual search computational models.

## Acknowledgements

The authors were supported by the National Science and Technology Research Council (CONICET, Argentina) and the University of Buenos Aires (UBA, Argentina). The research was supported by the CONICET (PIP 11220150100787CO), the ARL (Award W911NF1920240) and the National Agency of Promotion of Science and Technology (PICT 2018-2699). We thank P. Riera (UBA, Argentina), A. Zylberberg (Univ. of Rochester, US), A. Bendersky (UBA, Argentina), and E. Iarussi (Univ. Torcuato Di Tella, Argentina) for their feedback, help and insight on the work.

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
