# Appendix

## A    Methods and Materials

### A.1    Equalizing datasets

To bring each dataset to a common format, a series of steps were carried out. To begin with, all of the participants' scanpaths were cut as soon as a fixation had landed on the target and only trials in which the target was found were considered. The same criteria were applied to the models. In order to do this, an estimation of the size of the fovea (i.e. the region around the fixation point in which the target could be identified) had to be done for each dataset (Table 1). Consecutive fixations that were closer to each other than the size of the fovea were merged. With respect to the images, those in the *Unrestricted* dataset had to be transformed to grayscale, and the ones in the *MCS* dataset were rescaled to their average size. The targets' bounding boxes were not available for the latter, so they had to be retrieved from the COCO dataset's annotations, and target-absent trials were excluded.

Given the difference in the experiments' design, target templates representative of each object category had to be chosen from the web in both the *MCS* and *COCOSearch18* datasets (Fig. A1), for two of the models evaluated need a image of the target to search. This is similar to what the authors did in the experiment, where participants were shown a template of the target's category before starting each trial [28]. Conversely, images from the *Unrestricted* and *Interiors* datasets had to be categorized by hand in one of the COCO object categories depending on their target object, so they could be processed by the visual search model trained on the *COCOSearch18* dataset. These differences were also present in the participants' scanpaths length. Since every visual search model possesses a maximum number of fixations, this upper limit was set for each dataset according to the saturation of human performance.

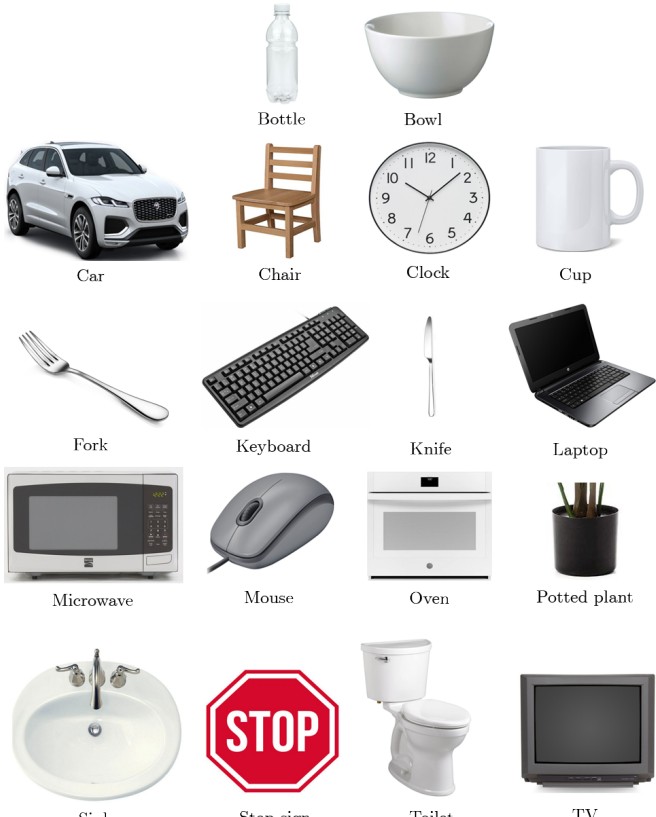

Figure A1: Templates used for each target category in the *COCOSearch18* and *MCS* datasets.

```
1    {
2    "dataset_name": [string] Dataset's name,
3    "number_of_images": [int] Total number of search images,
4    "image_height": [int] Height of the search images,
5    "image_width": [int] Width of the search images,
6    "max_scanpath_length": [int] Max. scanpath length allowed,
7    "receptive_size": [int, int] Fovea's size,
8    "mean_target_size": [int, int] Mean target size,
9    "images_dir": [string] Search images' path,
10   "targets_dir": [string] Targets' templates path,
11   "scanpaths_dir": [string] Participants' scanpaths path
12   }
```

Figure A2: Metadata for a given dataset. Size is specified as height and width.

```
1    [
2        {
3        "image": [string] Search image filename,
4        "target": [string] Target template filename,
5        "dataset": [string] Dataset's name,
6        "target_matched_row": [int] Target's upper left row in
             the search image,
7        "target_matched_column": [int] Target's upper left
             column in the search image,
8        "target_height": [int] Target's height,
9        "target_width": [int] Target's width,
10       "image_height": [int] Search image's height,
11       "image_width": [int] Search image's width,
12       "initial_fixation_row": [int] Row of the initial
             fixation,
13       "initial_fixation_column": [int] Column of the initial
             fixation,
14       "target_object": [string] COCO's category of the target
15       },
16       ...
17   ]
```

Figure A3: Metadata of each search image and target, loaded by the models.

Trivial search images (that is to say, those where the initial fixation lands on the target) were discarded in the *MCS* and *Unrestricted* datasets. A minority of the search images in the training set of the MCS dataset (around 6%) included two instances of the target category and were, thus, excluded. In the case of the *Unrestricted* dataset, images were shown twice to the participants, and here only the first trials were kept. *COCOSearch18* used some of the images for more than one trial, changing the target; these are regarded as different images.

Finally, a common format for participants' scanpaths and the input images and targets was defined, based on JSON files. The models' output follow the same format as those of participants. This way, a concordance in terms of nomenclature across all datasets and models is also achieved.

### A.1.1 Input format

Datasets have two main JSON files in their root directory: *datasets_info.json* (Fig. A2), which contains the datasets' metadata (such as the paths to the search images, targets and participants' scanpaths directories), and *trials_properties.json* (Fig. A4), where all the necessary information for running the models in each image is specified. Using this approach, visual search models only require the name of the dataset as input.

### A.1.2 Output format

Both participants' and models' scanpaths share the same format (Fig. A4), with the exception of the fields for the screen width and height (the resolution of the screen used during the experiments) and the scanpath's time dimension. The participants' scanpaths are located in the corresponding dataset's directory, while the models' scanpaths are saved as '*Scanpaths.json*' under the Results directory (subdivided by the datasets' names).

```
 1      {
 2      "image_name":{
 3              "subject": [string] Participant's ID,
 4              "dataset": [string] Dataset's name,
 5              "image_height": [int] Search image's height,
 6              "image_width": [int] Search image's width,
 7              "screen_height"(*): [int] Screen's resolution
                  height,
 8              "screen_width"(*): [int] Screen's resolution width,
 9              "receptive_height": [int] Height, in pixels, of the
                  fovea's size,
10              "receptive_width": [int] Width, in pixels, of the
                  fovea's size,
11              "target_found": [bool] True if the target was found
                  ,
12              "target_bbox": [array[int]] Target's bounding box
                  in the search image,
13              "X": [array[float]] Scanpath's column coordinates,
14              "Y": [array[float]] Scanpath's row coordinates,
15              "T"(*): [array[float]] Scanpath's time, in ms,
16              "target_object": [string] COCO's category of the
                  target,
17              "max_fixations": [int] Scanpath's length upper
                  bound
18          }
19          ...
20      }
```

Figure A4: Fields required for describing the scanpaths of each search image. Those marked with (*) only belong to the scanpaths of participants. The target bounding box is specified as the upper left and bottom right vertices in the search image, where each vertex is in height and width notation.

### A.2 Models' scanpath parameters

Despite the differences between models, we attempt to set a common criterion for deciding when the target was found. All of them use an oracle to decide if the target was found, and the search is carried out until the target is found or a maximum number of fixations is reached. It is worth noting that it is not possible to apply precisely the same rule to every one of them, since IVSN runs over the original image and the IBS family and IRL run over different grids (Table A1).

### A.3 IBS Python implementation

The original code, written in MATLAB, was ported to Python, an open-source programming language available to everyone. From the start, the goal in mind was to be as clear as possible when it came to writing the code, including a precise definition of the input and output of each method. Additionally, we focused on an object-oriented approach, creating classes for each important concept within the model (prior, visibility map, target similarity map, and so on). All of this allows for different hypotheses to be tested with ease, including the resulting models in the present work, and future improvements (such as modifying the prior, in light of the results displayed here) can be implemented without much trouble. We believe well-written code is central for reproducible research.

Table A1: Description of how each model regards that the target was found.

| Model | Target Bounding Box | Fovea | End of search criteria |
|-------|--------------------|-------|------------------------|
| IBS | Square centered in the target. Projected to grid. | 1x1 in 24x32 grid | Fovea lands in bounding box |
| IVSN | Surrounds the target in original image size. Rectangular. | Mean target size in original image size | Fovea lands in bounding box |
| IRL | Surrounds the target in 320x512. Rectangular. | 1x1 in 20x32 grid | Fovea lands in bounding box (rescaled to 320x512) |

Even though the model is an exact port of its MATLAB implementation, there remain differences in the generation of pseudo-random noise, which impacts the creation of scanpaths, for the algorithm utilized in MATLAB is not disclosed and, hence, can not be reproduced. Notwithstanding, these differences do not impact the model's performance. On the other hand, since the model's hyper-parameters are tuned for images whose resolution is 768x1024, its input is automatically resized to this size, where the visibility and target similarity maps are computed, before downscaling it to a grid.

### A.3.1 Prior estimation

The estimation of the prior is done by computing saliency maps using the center bias over the input images, which in this case is done via DeepGaze II [33]. In order to run the model on several different datasets, the estimation of the prior is now incorporated into the model through a TensorFlow adaptation of a Jupyter Notebook provided by the BETHGE LAB[6].

### A.3.2 Cross-correlation in colored images

Given that the input of the original model was grayscale images, an adaptation had to be done in order to run the model on datasets where images are colored. Cross-correlation returns a different value for each color channel, so a weighted-average was calculated from common RGB to grayscale formulas. In particular, we chose the one used by the `scikit-image` library:

$$Y = 0.2125 * R + 0.7154 * G + 0.0721 * B$$

### A.3.3 SSIM

Due to the extensive use of SSIM in the video industry (to quantify for image quality, for instance), it arose as a natural replacement for cross-correlation when it came to computing the similarity between the search image and target. Moreover, it is capable of working in colored images.

SSIM is computed on the target template and each possible region of the image. Then, these results are averaged and stored in the center pixel of the corresponding region. With this approach, black bars would remain at the edges of the images, so they are managed differently: we trim from the target template those pixels that would overflow the image's bounds and then we compute SSIM between the template's leftovers and the corresponding region of the image. Since these calculations are made pixel by pixel, the computation behind this method is very expensive, but they have to be done only once (for the result of each target template and search image is stored and can be reutilized).

### A.3.4 Neural network

By making use of our PyTorch implementation of the IVSN model, it became possible to adapt the attention map computed by IVSN [22]. Since its values range from zero to one, this adaptation was pretty straightforward: it simply serves as a replacement for cross-correlation. The advantages of capturing object invariance while retaining a good performance in searching for templates with little cost in computation made it our best choice for this benchmark.

---

[6]`https://deepgaze.bethgelab.org`

### A.4 IVSN Python implementation

The code available[7] is written in both MATLAB and LUA, using the first for pre and postprocessing and the latter for computations with the Caffe VGG16 model. Besides making the necessary adjustments to read the input and write the output in the previously defined JSON format, our Python implementation uses the PyTorch library alongside its implementation of VGG16.

Given the greedy nature of the algorithm, changes on the scanpaths produced were observed depending on small factors, such as using a different formula for the grayscale conversion of the input images or a different interpolation method for upscaling the neural network's output. Nonetheless, its performance remained unaffected (Fig. A5)

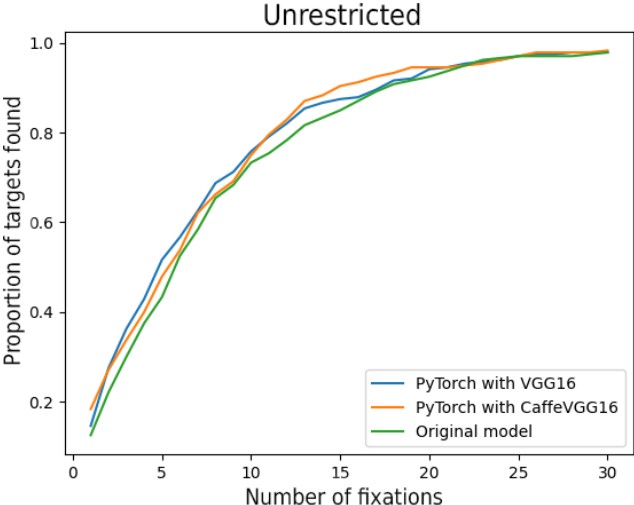

Figure A5: Performance comparison between the different versions of IVSN, in its own dataset, with the original target found criteria defined by the authors.

### A.5 Adapting the IRL model

This model is already written in Python,[8] but its execution is not straightforward, since only the code for training the neural network is available. Moreover, the preprocessing of the search images described in [24] was not specified in the code.

Following their description, in our implementation, input images are resized to 320x512 and a gaussian blur with $\sigma = 2$ is applied to them, producing two images for each search image. Both of these are fed forward through the Panoptic-FPN in Detectron2 [36] with Resnet-50 3X as backbone[9] and the output is resized to 20x32. We found differences in IRL's performance depending on how this downscaling was done. Detectron2 itself provides an output size as a parameter, but the performance was meager when compared to downscaling using common image interpolation techniques (such as nearest-neighbor), so the latter was used in our implementation. In all cases, we adopt the criteria of using the best performing model when some specifications are missing.

Once this has been done, masks are created from these semantic segmentations for each of the 134 COCO categories (if a category was not present in the search image, then its corresponding mask is simply the null matrix). These masks are the model's input.

All of this preprocessing has been integrated to the model itself, so our implementation is able to create the so-called Dynamic Context Belief maps on any given image. However, the categorization of the target object has to be done by hand.

---

[7]https://github.com/kreimanlab/VisualSearchZeroShot
[8]https://github.com/cvlab-stonybrook/Scanpath_Prediction
[9]https://github.com/facebookresearch/detectron2

## A.6 Human-model and within-human Multi-Match

Given the models' scanpaths and those of participants, it is possible to do a pairwise comparison employing Multi-Match and then compute the average across participants, obtaining what we call human-model Multi-Match (hmMM) and, when comparing scanpaths within participants, within-human Multi-Match (whMM).

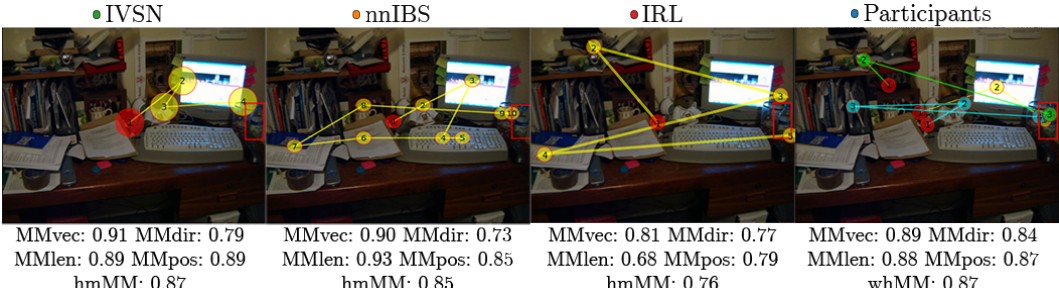

| MMvec: 0.91 MMdir: 0.79 | MMvec: 0.90 MMdir: 0.73 | MMvec: 0.81 MMdir: 0.77 | MMvec: 0.89 MMdir: 0.84 |
| MMlen: 0.89 MMpos: 0.89 | MMlen: 0.93 MMpos: 0.85 | MMlen: 0.68 MMpos: 0.79 | MMlen: 0.88 MMpos: 0.87 |
| hmMM: 0.87 | hmMM: 0.85 | hmMM: 0.76 | whMM: 0.87 |

Figure A6: Multi-Match mean values in every dimension (shape, direction, length, and position), with the exception of the temporal one, and their average.

## A.7 Human Scanpath Prediction (HSP)

By forcing each model to follow the participants' scanpaths, it is possible to compute several established metrics based on the (conditional) priority map they compute to sample the next fixation and assess how well their prediction of the next human fixation was. Using the conditional priority map as a binary classifier (classifying locations into fixated and non-fixated), AUC*hsp* measures the rank of the fixated pixel, where an AUC of 100% corresponds to a perfect prediction and an AUC of 50% corresponds to chance level (all pixels are considered as non-fixations). NSS*hsp* simply computes the average saliency of the fixated location after normalizing the conditional priority map to have zero mean and unit variance [30]. To avoid numeric errors, any number below the threshold of $1 * 10^{-20}$ is regarded as 0. Finally, information gain is calculated as the difference in average log-likelihood between the visual search model and a baseline model [40]; here, IG*hsp* is relative to the center bias model, while LL*hsp* is relative to the uniform model, using logarithms with base 2 [30].

### A.7.1 Procedure

The computation of the conditional priority maps is achieved by modifying the models' input with an optional parameter that corresponds to the ID of the participant on which to run the metrics. Each model is responsible for loading the participant's scanpaths and following them, with the modifications described in 2.3. In a given scanpath, for each fixation $n$, with $n$ greater than zero (the initial fixation is fixation zero), the conditional priority map is saved under the name *'fixation_n.csv'* in the same directory where the model's results for that dataset reside. Once this is done for the whole scanpath, an external method is called which proceeds to compute the metrics for each fixation and stores the average.

### A.8 Resources and computation

For the computation of the previously stated models and metrics, using the described datasets as input, one main PC was used locally (without accessing any GPUs): an AMD Ryzen 5 2600X, six cores at 3600MHz, processor with 16GB dual-channel RAM DDR4 at 2133MHz.

The time complexity of the models is dominated by the IBS family models, as both IRL and IVSN are pre-trained neural networks on which data are forwarded. Moreover, there is a significant overhead when running sIBS in comparison to the other two variations (cIBS and nnIBS), due to the calculations required for calculating SSIM (see A.3.3). In this case, the computation of the target similarity map of a single image can take up to eight minutes, totalling over 80 hours in a dataset

such as *COCOSearch18* (which contains 612 images). In these three models, the computation of a saccade takes about 5 seconds, totalling near 5 hours in *COCOSearch18*.

The metrics based solely on scanpaths (cumulative performance and Multi-Match) take less than a minute per dataset. On the other hand, following a participant's scanpaths (human scanpath prediction) requires the complete execution of the models on each dataset the same number of times as the number of participants (in the case of IBS and *COCOSearch18*, that amounts to over 40 hours).

# B   Results

## B.1   Comparison of different similarity maps within the IBS family

Results for the different methods used in the target similarity map computation (Table A2; Table A3): cross-correlation (cIBS), SSIM (sIBS), and neural network-based (nnIBS).

Table A2: IBS performance (AUC*perf*), the average scanpath similarity metrics (AvgMM) and its correlation against humans (Corr, values not significantly different from 0 ($pval > 0.05$) are marked as (*)), and human scanpath prediction metrics (HSP), as defined in 2.3.

|  | AUC*perf* | AvgMM | Corr | AUC*hsp* | NSS*hsp* | IG*hsp* | LL*hsp* | Score |
|---|---|---|---|---|---|---|---|---|
| **Interiors** | | | | | | | | |
| • Humans | 0.42 | 0.86 | - | - | - | - | - | - |
| Gold Standard | - | - | - | 0.86 | 1.87 | 1.83 | 1.43 | 0.00 |
| • nnIBS | 0.53 | **0.84** | **0.28** | **0.70** | **0.94** | **0.92** | **0.23** | -0.12 |
| • cIBS | **0.52** | **0.84** | 0.25 | 0.69 | **0.94** | 0.88 | 0.18 | -0.13 |
| • sIBS | 0.56 | 0.83 | 0.11* | 0.69 | 0.93 | 0.87 | 0.18 | -0.16 |
| Center bias | - | - | - | 0.67 | 0.71 | 0.00 | -0.41 | -0.78 |
| Uniform | - | - | - | 0.50 | 0.00 | 0.41 | 0.00 | -0.80 |
| **Unrestricted** | | | | | | | | |
| • Humans | 0.53 | 0.86 | - | - | - | - | - | - |
| Gold Standard | - | - | - | 0.86 | 2.03 | 1.70 | 1.51 | 0.00 |
| • nnIBS | 0.63 | 0.82 | 0.01* | 0.72 | **1.06** | **0.76** | **0.31** | -0.16 |
| • cIBS | **0.54** | **0.83** | 0.05* | **0.74** | **1.06** | 0.62 | 0.18 | -0.16 |
| • sIBS | 0.61 | 0.83 | **0.14** | 0.72 | 1.05 | 0.62 | 0.18 | -0.16 |
| Center bias | - | - | - | 0.70 | 0.71 | 0.00 | -0.19 | -0.74 |
| Uniform | - | - | - | 0.50 | 0.00 | 0.19 | 0.00 | -0.83 |
| **COCOSearch18** | | | | | | | | |
| • Humans | 0.80 | 0.88 | - | - | - | - | - | - |
| Gold Standard | - | - | - | 0.96 | 4.20 | 3.17 | 3.10 | 0.00 |
| • nnIBS | **0.52** | **0.85** | 0.14 | 0.75 | **1.08** | **0.46** | **0.32** | -0.27 |
| • cIBS | 0.47 | **0.85** | **0.17** | **0.77** | **1.08** | 0.40 | 0.26 | -0.28 |
| • sIBS | 0.48 | **0.85** | 0.13 | 0.76 | 1.07 | 0.40 | 0.27 | -0.28 |
| Center bias | - | - | - | 0.72 | 0.52 | 0.00 | -0.07 | -0.79 |
| Uniform | - | - | - | 0.5 | 0.00 | 0.07 | 0.00 | -0.86 |
| **MCS** | | | | | | | | |
| • Humans | 0.49 | 0.89 | - | - | - | - | - | - |
| Gold Standard | - | - | - | 0.91 | 2.50 | 1.03 | 1.77 | 0.00 |
| • nnIBS | 0.51 | 0.86 | 0.15 | 0.79 | **1.98** | **-0.39** | 0.55 | -0.19 |
| • cIBS | **0.50** | **0.87** | **0.20** | **0.80** | 1.96 | -0.65 | 0.29 | -0.23 |
| • sIBS | 0.51 | 0.86 | 0.16 | 0.79 | 1.94 | -0.65 | 0.29 | -0.25 |
| Center bias | - | - | - | 0.81 | 1.65 | 0.00 | **0.95** | -0.48 |
| Uniform | - | - | - | 0.5 | 0.00 | -0.95 | 0.00 | -1.09 |

Table A3: IBS average scanpath similarity metrics (AvgMM) and all its dimensions (shape, direction, length and position) (± standard error).

|  | AvgMM | MMvec | MMdir | MMlen | MMpos |
|---|---|---|---|---|---|
| **Interiors** | | | | | |
| • Humans | 0.85 | 0.93±0.01 | 0.73±0.05 | 0.92±0.02 | 0.84±0.03 |
| • nnIBS | 0.84 | 0.91±0.02 | 0.71±0.06 | 0.90±0.03 | 0.84±0.04 |
| • cIBS | 0.84 | 0.91±0.02 | 0.71±0.06 | 0.90±0.03 | 0.83±0.04 |
| • sIBS | 0.84 | 0.90±0.02 | 0.71±0.06 | 0.89±0.04 | 0.83±0.04 |
| **Unrestricted** | | | | | |
| • Humans | 0.86 | 0.93±0.01 | 0.72±0.05 | 0.93±0.02 | 0.85±0.05 |
| • nnIBS | 0.82 | 0.91±0.02 | 0.67±0.07 | 0.89±0.04 | 0.83±0.04 |
| • cIBS | 0.83 | 0.92±0.02 | 0.67±0.08 | 0.91±0.03 | 0.83±0.04 |
| • sIBS | 0.83 | 0.91±0.02 | 0.68±0.07 | 0.90±0.03 | 0.83±0.04 |
| **COCOSearch18** | | | | | |
| • Humans | 0.88 | 0.93±0.03 | 0.76±0.11 | 0.92±0.04 | 0.90±0.04 |
| • nnIBS | 0.85 | 0.91±0.03 | 0.69±0.12 | 0.91±0.04 | 0.88±0.04 |
| • cIBS | 0.85 | 0.92±0.03 | 0.68±0.12 | 0.91±0.04 | 0.88±0.04 |
| • sIBS | 0.85 | 0.92±0.03 | 0.68±0.13 | 0.91±0.04 | 0.88±0.04 |
| **MCS** | | | | | |
| • Humans | 0.89 | 0.94±0.02 | 0.75±0.11 | 0.93±0.04 | 0.92±0.04 |
| • nnIBS | 0.86 | 0.92±0.03 | 0.71±0.13 | 0.91±0.05 | 0.90±0.05 |
| • cIBS | 0.87 | 0.93±0.03 | 0.72±0.12 | 0.92±0.04 | 0.90±0.05 |
| • sIBS | 0.86 | 0.92±0.03 | 0.71±0.13 | 0.91±0.05 | 0.90±0.04 |

### B.1.1 MCS dataset plot

Due to size constraints, the plot results for the IBS family models in the *MCS* dataset are displayed in Figure A7.

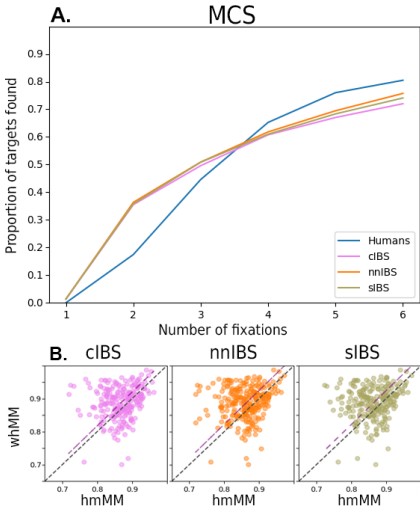

Figure A7: **A)** Ratio of targets found (vertical axis) for a given number of fixations (horizontal axis). **B)** Mean MM values within humans (whMM) as a function of the mean MM values between each human observer and a model (hmMM). Each dot represents an image.

## B.2 Comparison of different patch sizes for the IVSN model

The behavior of IVSN is deeply ingrained with the patch size it uses for target identification and to apply the inhibition-of-return effect while searching in its attention map (Fig. A8).

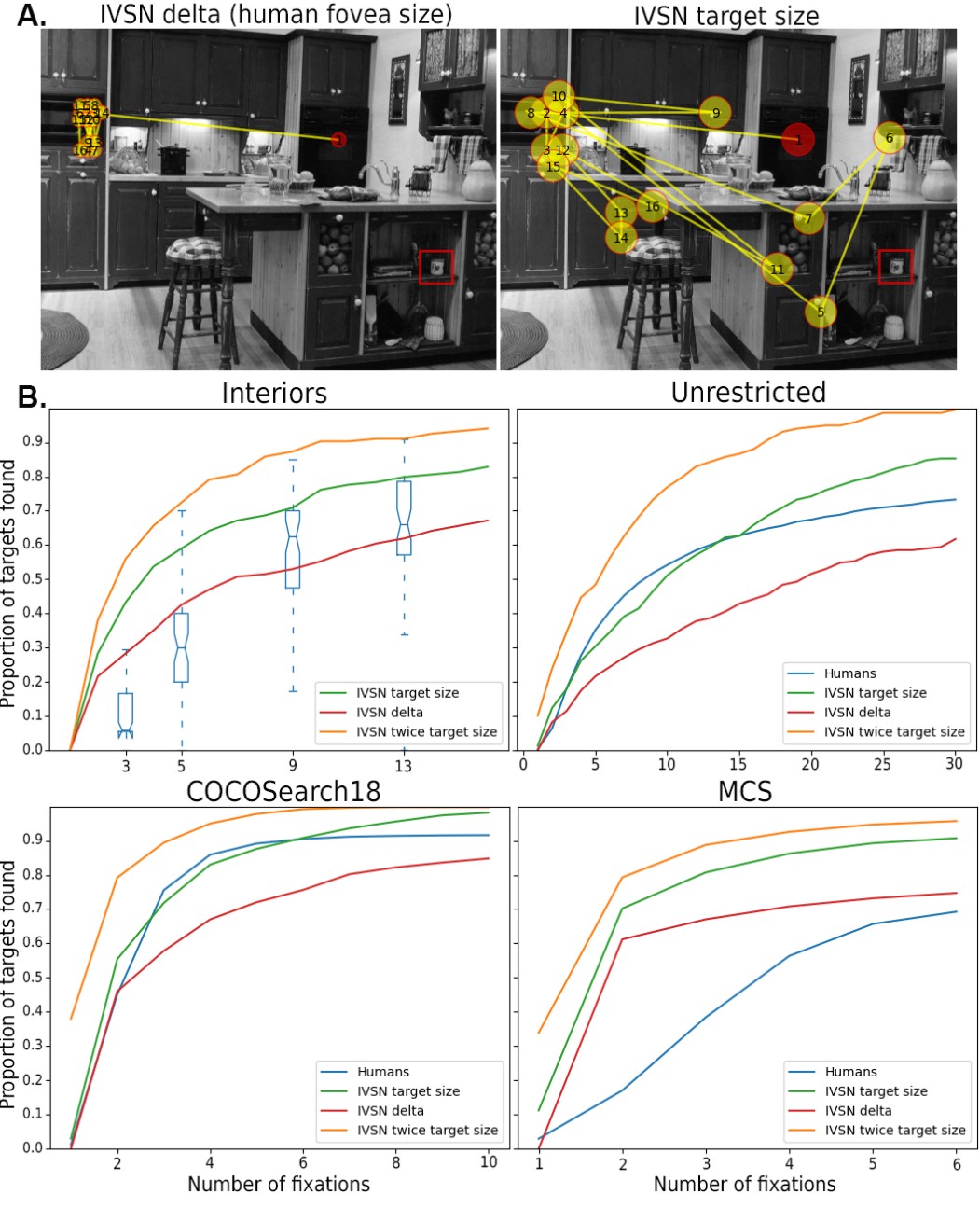

Figure A8: Three different patch sizes are tested for IVSN: the size of the human fovea (*delta*, in red), the mean target size (*target size*, in green) or twice the mean target size (original criteria used by Zhang et. al., in orange). **A)** A small patch size (left) causes the greedy algorithm to get stuck. **B)** Cumulative performance in all datasets of the different patch sizes tested.

## B.3 Comparison between modeling approaches

Table A4: Models' performance (AUC*perf*), the average scanpath similarity metrics (AvgMM) and its correlation against humans (Corr, values not significantly different from 0 ($pval > 0.05$) are marked as (*)), and human scanpath prediction metrics (HSP), as defined in 2.3.

| | AUC*perf* | AvgMM | Corr | AUC*hsp* | NSS*hsp* | IG*hsp* | LL*hsp* | Score |
|---|---|---|---|---|---|---|---|---|
| **Interiors** | | | | | | | | |
| • Humans | 0.42 | 0.86 | - | - | - | - | - | - |
| Gold Standard | - | - | - | 0.86 | 1.87 | 1.83 | 1.43 | 0.00 |
| • nnIBS | **0.53** | **0.84** | **0.28** | **0.70** | **0.94** | **0.92** | **0.23** | -0.12 |
| • IVSN | 0.65 | 0.79 | 0.23* | 0.59 | 0.76 | -2.19 | -2.58 | -0.71 |
| Center bias | - | - | - | 0.66 | 0.71 | 0.00 | -0.41 | -0.78 |
| Uniform | - | - | - | 0.50 | 0.00 | 0.41 | 0.00 | -0.80 |
| • IRL | 0.24 | 0.79 | -0.10* | 0.58 | 0.52 | -4.90 | -5.56 | -1.28 |
| **Unrestricted** | | | | | | | | |
| • Humans | 0.53 | 0.86 | - | - | - | - | - | - |
| Gold Standard | - | - | - | 0.86 | 2.03 | 1.70 | 1.51 | 0.00 |
| • nnIBS | 0.63 | **0.82** | 0.01* | **0.72** | **1.06** | **0.76** | **0.31** | -0.16 |
| Center bias | - | - | - | 0.70 | 0.71 | 0.00 | -0.19 | -0.74 |
| Uniform | - | - | - | 0.50 | 0.00 | 0.19 | 0.00 | -0.83 |
| • IVSN | **0.56** | 0.79 | 0.05* | 0.59 | 0.71 | -3.31 | -3.44 | -0.89 |
| • IRL | 0.23 | 0.80 | 0.26* | 0.58 | 0.50 | -5.91 | -6.36 | -1.41 |
| **COCOSearch18** | | | | | | | | |
| • Humans | 0.80 | 0.88 | - | - | - | - | - | - |
| Gold Standard | - | - | - | 0.96 | 4.20 | 3.17 | 3.10 | 0.00 |
| • nnIBS | 0.52 | **0.85** | **0.14** | 0.75 | 1.08 | **0.46** | **0.32** | -0.27 |
| • IRL | **0.73** | 0.80 | 0.07* | **0.80** | **2.57** | -1.40 | -1.56 | -0.37 |
| • IVSN | **0.73** | 0.81 | 0.11* | 0.64 | 1.13 | -3.68 | -3.73 | -0.64 |
| Center bias | - | - | - | 0.72 | 0.52 | 0.00 | -0.07 | -0.79 |
| Uniform | - | - | - | 0.5 | 0.00 | 0.07 | 0.00 | -0.86 |
| **MCS** | | | | | | | | |
| • Humans | 0.49 | 0.89 | - | - | - | - | - | - |
| Gold Standard | - | - | - | 0.91 | 2.50 | 1.03 | 1.77 | 0.00 |
| • nnIBS | **0.51** | **0.86** | **0.15** | 0.79 | **1.98** | **-0.39** | 0.55 | -0.19 |
| Center bias | - | - | - | **0.81** | 1.65 | 0.00 | **0.95** | -0.48 |
| Uniform | - | - | - | 0.5 | 0.00 | -0.95 | 0.00 | -1.09 |
| • IRL | 0.38 | 0.80 | -0.08* | 0.64 | 1.39 | -7.10 | -6.13 | -1.77 |
| • IVSN | 0.76 | 0.82 | -0.004* | 0.61 | 1.68 | -7.96 | -6.96 | -1.96 |

## B.4 Comparison between modeling approaches: some failure examples

Here, we present examples which illustrate some failures of the models. In every figure, initial fixations are displayed in red, while participants' scanpaths are superimposed with different colors.

In Figure A9, two different examples are shown where IVSN fell for distractors similar to the target, whereas nnIBS proved to be more robust. Figure A10 depicts two cases in which there is a sheer difference between IVSN and participants in terms of saccade amplitude, highlighting the relevance of modeling visual decay in the periphery. Figure A11 shows how participants are capable of incorporating both the context of the image and the target's meaning to guide their search accordingly, a behavior that remains elusive for current visual search models. Finally, in Figure A12 participants' and models' scanpaths on images where human faces were present are displayed. Even though participants didn't fixate on them, nnIBS did, mainly in the beginning, due to its prior.

Table A5: Visual search models average scanpath similarity metrics (AvgMM) and all its dimensions (shape, direction, length and position) (± standard error).

| | AvgMM | MMvec | MMdir | MMlen | MMpos |
|---|---|---|---|---|---|
| **Interiors** | | | | | |
| • Humans | 0.85 | 0.93±0.01 | 0.73±0.05 | 0.92±0.02 | 0.84±0.03 |
| • nnIBS | **0.84** | **0.91**±0.02 | **0.72**±0.06 | **0.90**±0.03 | **0.84**±0.03 |
| • IVSN | 0.79 | 0.87±0.05 | 0.69±0.08 | 0.83±0.10 | 0.79±0.07 |
| • IRL | 0.80 | 0.89±0.04 | 0.65±0.06 | 0.85±0.06 | 0.79±0.05 |
| **Unrestricted** | | | | | |
| • Humans | 0.86 | 0.94±0.01 | 0.72±0.05 | 0.93±0.02 | 0.85±0.05 |
| • nnIBS | **0.82** | **0.91**±0.02 | 0.67±0.07 | **0.89**±0.04 | **0.83**±0.04 |
| • IVSN | 0.79 | 0.88±0.04 | 0.64±0.07 | 0.85±0.07 | 0.78±0.06 |
| • IRL | 0.80 | 0.88±0.04 | **0.68**±0.06 | 0.86±0.07 | 0.79±0.05 |
| **COCOSearch18** | | | | | |
| • Humans | 0.88 | 0.93±0.03 | 0.77±0.11 | 0.92±0.04 | 0.90±0.04 |
| • nnIBS | **0.85** | **0.91**±0.03 | **0.69**±0.01 | **0.91**±0.04 | **0.88**±0.04 |
| • IVSN | 0.80 | 0.88±0.05 | 0.65±0.12 | 0.85±0.07 | 0.81±0.08 |
| • IRL | 0.81 | 0.88±0.04 | 0.68±0.12 | 0.85±0.07 | 0.82±0.07 |
| **MCS** | | | | | |
| • Humans | 0.89 | 0.95±0.02 | 0.75±0.11 | 0.93±0.04 | 0.92±0.04 |
| • nnIBS | **0.86** | **0.93**±0.03 | **0.71**±0.13 | **0.91**±0.05 | **0.90**±0.05 |
| • IVSN | 0.80 | 0.88±0.04 | 0.65±0.12 | 0.84±0.07 | 0.81±0.07 |
| • IRL | 0.82 | 0.89±0.04 | 0.67±0.14 | 0.87±0.07 | 0.86±0.06 |

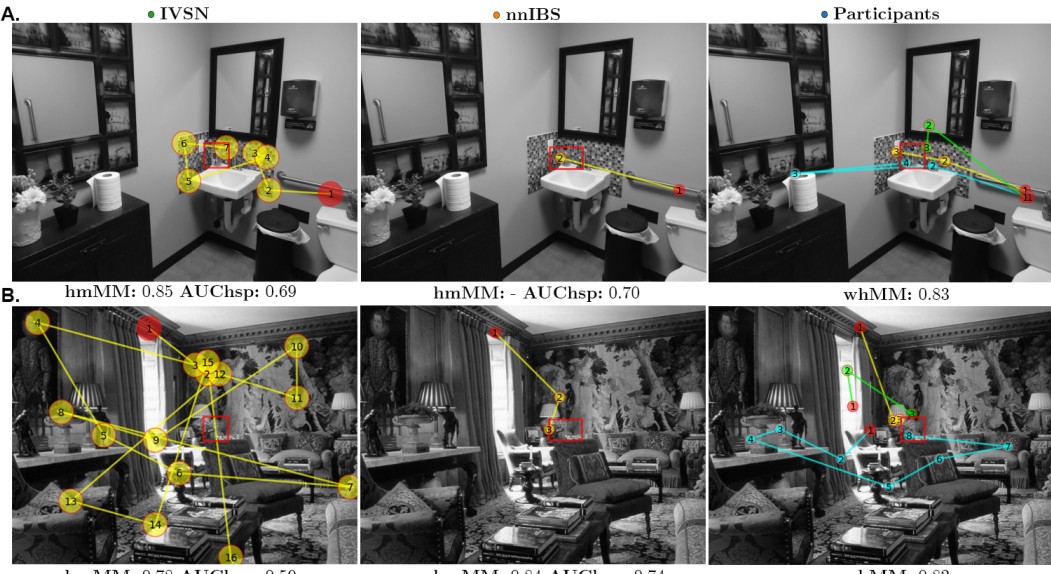

Figure A9: Even though nnIBS and IVSN share their target similarity map, IVSN falls for distractors due to its greedy algorithm. **A)** The pattern on the wall fools IVSN. MM could not be computed on nnIBS's scanpath due to its short length. **B)** While looking for a bust, IVSN fixates on faces in paintings (fixation two, three and four). The targets in these images do not fall under the allowed target categories for IRL, so its scanpaths could not be displayed.

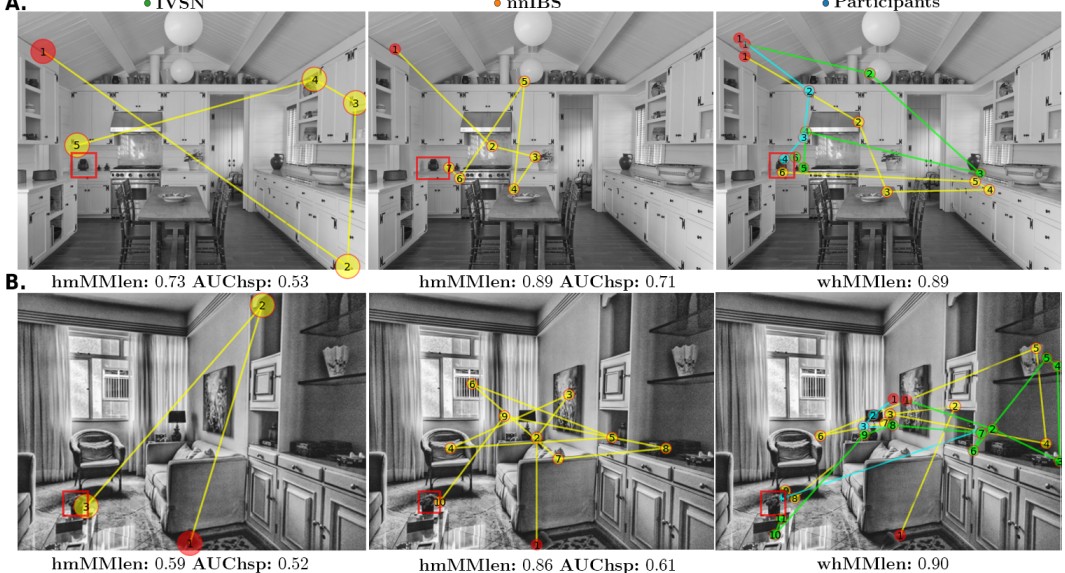

Figure A10: IVSN, due to its lack of a model of the fovea, crosses the entire image in one saccade. The targets in these images do not fall under the allowed target categories for IRL, so its scanpaths could not be displayed

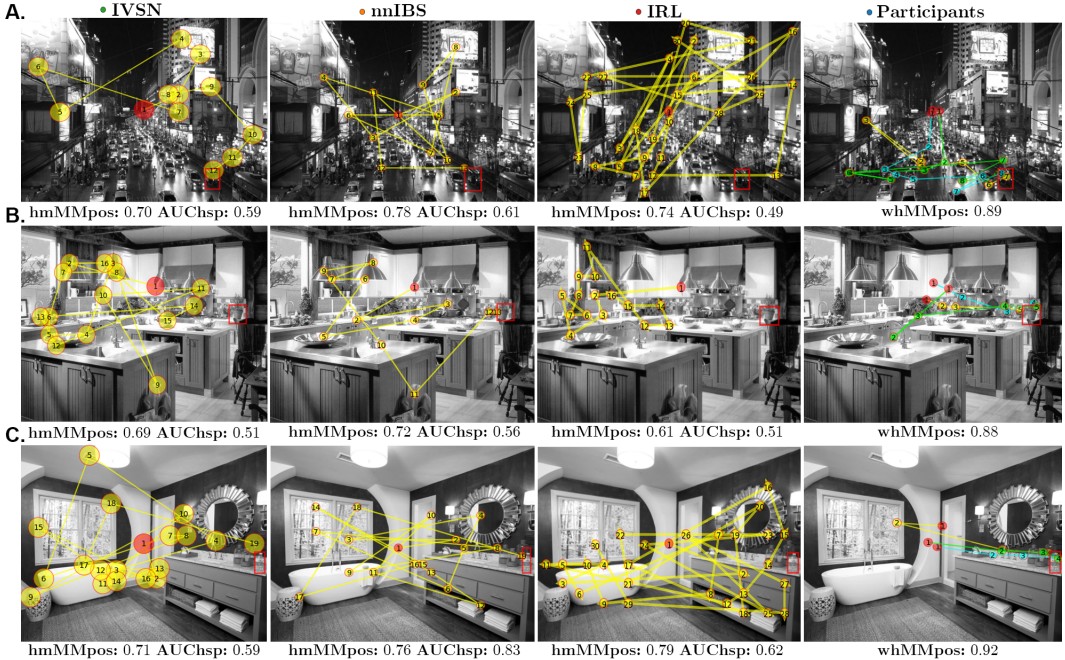

Figure A11: Search images where between-human Multi-Match was greatest and model-humans Multi-Match was lowest, showing its position dimension. **A)** Participants immediately fixated on the road while looking for a car. **B)** The target is a potted plant, so participants only looked at the kitchen counter level. **C)** Similar to B), where the target is a bottle placed in a bathroom.

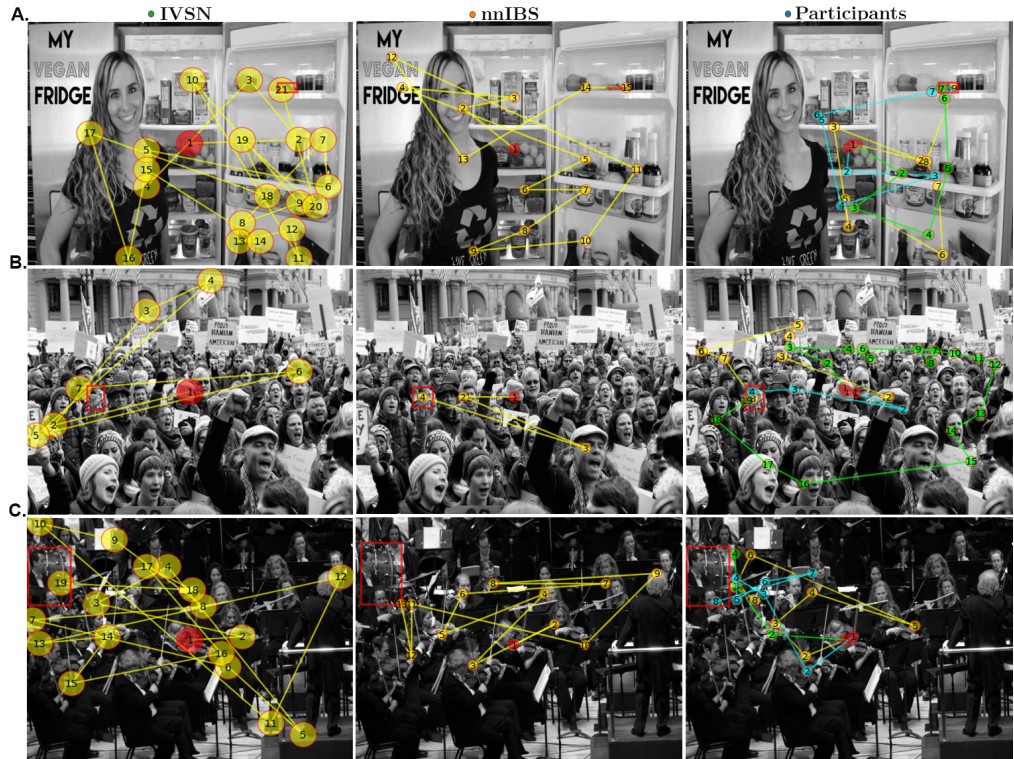

Figure A12: nnIBS, regardless of the target, fixates on human faces in the first few fixations. This pattern does not occur in human subjects. The targets in these images do not fall under the allowed target categories for IRL, so its scanpaths could not be displayed.

## C   Licenses

The *Interiors* dataset, the cIBS, sIBS and IRL models were released under the MIT License [10][11]. The *Unrestricted* dataset and the IVSN model were released under Kreiman lab's license agreement [12]. The *COCOSearch18* and *MCS* datasets were released for public use with restrictions stated on their official website [13][14]. And the Detectron2 model was released under the Apache 2.0 license [15].

---

[10] https://github.com/gastonbujia/VisualSearch/blob/master/license
[11] https://github.com/cvlab-stonybrook/Scanpath_Prediction/blob/master/LICENSE
[12] https://klab.tch.harvard.edu/code/license_agreement.pdf
[13] https://sites.google.com/view/cocosearch/
[14] https://sites.google.com/view/mcs-dataset/home
[15] https://github.com/facebookresearch/detectron2/blob/main/LICENSE