# OpenReview forum: "ViSioNS: Visual Search in Natural Scenes Benchmark"
_NeurIPS.cc/2022/Track/Datasets_and_Benchmarks — NeurIPS 2022 Datasets and Benchmarks _

### Official Review · Reviewer_unJU · 2022-07-20
**Interesting, but the novel contribution is not clear**

**Rating:** 6
**Confidence:** 4
**Clarity:** Yes.

**Strengths:**

1. The paper presents a very thorough and detailed comparison between the existing datasets and models in the task of visual search. The key differences are made clear and the lack of uniformity and standardization is made readily apparent.

2. The results presented in the paper highlight the room for improvement in the task of visual search computation and seem to be reproducible. The work clearly shows that each method considered has benefits and drawbacks, and does a good job of providing a reasonable explanation and interpretation of the results.

**Weaknesses:**

While this paper does a good job of reviewing the current methods in visual search and even providing quantitative results of existing models  on existing datasets evaluated using existing metrics, the paper does not provide a better alternative for unifying the evaluation procedure. In other words, while the paper convinces the reader that there is a need to unify the criteria used to evaluate visual search methods, it does not provide such a unified criteria. A more significant novel contribution could come in the form of a metric that has a generalizable ability to compare performance across different datasets and models.

The only novel contribution seems to be applying existing models to existing datasets and evaluating them using existing metrics. In this way, this paper seems to be more suited as a review paper rather than a benchmark paper. All of the results and analyses presented in this paper seem to be correct. However, while they are interesting, they do not constitute a significant enough contribution to the effort of benchmarking model performance in visual search.

**Additional Feedback:**

I think that while the paper provides a fantastic overview of visual search methods, it lacks a significant novel contribution. While it is made clear that there is no unified criteria for evaluation, the paper would be much more convincing if it provided an alternative.

**Correctness:**

Yes. The benchmark results seem to be correct to the best of my knowledge. Code is provided so that the results are reproducible.

**Documentation:**

Yes.

**Ethics:**

No.

**Relation To Prior Work:**

It would be beneficial to mention any other reviews of existing visual search methods. If none exist, it would be helpful to mention that.

**Summary And Contributions:**

The paper summarizes the various datasets and methods currently used in the prediction of human gaze positions during observation of images. First, it presents the many different datasets used for this task and highlights the key differences between each of the datasets. Namely, there is large variability in the number of participants surveyed, total number of images, size of the targets, format of the targets, stopping conditions of gaze tracking, and other attributes. Then, the different visual search models that are currently used are presented, as well as the metrics used to evaluate the performance of these models.

Many different experiments are performed with these datasets, models, and metrics. The authors conclude that there is still a large performance gap between these existing methods and the "Gold Standard" baseline model. The paper is concluded with a call for a more unified set of data and evaluation metrics for the task of visual search computation.

---

### Official Review · Reviewer_ren7 · 2022-07-23
**ViSioNS: Visual Search in Natural Scenes Benchmark**

**Rating:** 5
**Confidence:** 3
**Clarity:** Yes, it's understandable.

**Strengths:**

(+) code is open-sourced.

(+) The experiments conducted on the four datasets are comprehensive.

**Weaknesses:**

(-) I have a hard time finding contributions presented by this work, aside from running these baseline models on these datasets.
It would be nice if the authors can list their contributions in a bullet point fashion.

(-) Why is it that only 4 datasets are used? The author said that it is essential to always evaluate the visual search models in a diverse collection of datasets but 4 does not seem to be enough. Simulated datasets could also be considered right?

**Additional Feedback:**

above.

**Correctness:**

It seems to me that the dataset and baseline models are designed in a reasonable way.

**Documentation:**

Yes. They open-sourced their code.



**Ethics:**

I think it's fine.

**Relation To Prior Work:**

I think the contribution is weak because it is simply an aggregation of a couple of prior works.

**Summary And Contributions:**

They compared state-of-the-art visual search models in natural scenes. They benchmarked three publicly available models and defined a common set of metrics and baseline models and bring them together into a single pipeline.

---

### Official Review · Reviewer_HFYJ · 2022-07-24
**Benchmark for visual search tasks**

**Rating:** 7
**Confidence:** 4

**Strengths:**

This submission raised an important challenge, which is how to evaluate model performance on human gaze prediction. The diversity of tasks involving human subjects make it challenging to compare model performance if each model uses its own dataset. For example, the scanpath for a a goal driven visual search task would be drastically different from a free exploring task. I think creating a common dataset as proposed here has great value to the broader community and have potential for both computer vision and neuroscience.

**Weaknesses:**

There is one major limitation for this submission as a benchmark paper. As specifically pointed out in the paper, the existing datasets for human gaze prediction are for different purposes with different experimental designs. The submission tried their best to pre-process the data for common features. However, different datasets are still created under different experimental constraints.  It would be ideal if there exist a dataset for the benchmarking purpose which takes all the issues the paper raised into account.

**Additional Feedback:**

Additional tutorials that can help incorporate other models or preprocess other datasets would be helpful.

**Clarity:**

The paper is clearly written and well-organized. All methods and terminologies are clearly explained.

**Correctness:**

The paper is constructed in a sound way. The evaluation metrics are well explained and performed for testing.

**Documentation:**

The github repo is clearly organized. It can be better by providing generalized code to include other models and datasets.

**Ethics:**

No.

**Relation To Prior Work:**

This paper has clearly summarized the issue of previously collected datasets for benchmarking visual search task and provided a possible solution.

**Summary And Contributions:**

This submission aims to provide a benchmark for visual search behavior of natural scene images and a set of metrics and common pipeline to evaluate model performance. Saccade is a natural behavior in human vision and gaze position prediction is a major component of computer vision. Having a standard benchmark dataset to evaluate the gaze predictability across models has great value to the community.

This submission evaluated three publicly available models of human visual search behavior with preprocessed public datasets, and provided a common framework for model evaluation. The submission contains a well-organized github repo which clearly summarized preprocessed human scanpaths data from four different publicly available datasets, metrics (AUC, MultiMatch, Human Scanpath Prediction), example models to be tested and testing results.

---

### Official Review · Reviewer_B3vx · 2022-07-27
**A valuable framework for comparison of disparate models and datasets**

**Rating:** 7
**Confidence:** 2

**Strengths:**

The paper is directly and broadly relevant to the domain of visual search, and provides a framework in which analysis of models can be be performed, which is, in itself a valuable contribution to the field. The analysis provided covers a range of metrics which attempt to give a full picture of the similarities in scanpaths in a range of different contexts, and which are ultimately summarised into a single metric for the comparison of human and model-based sanpaths. The authors provide a codebase which makes a number of different models and datasets available within a common framework - making the analysis very accessible for others to build on.
Given that the analysis provided is of existing datasets and models, I don't see any obvious ethical concerns. Questions of subject consent should have been dealt with by the original authors of the published datasets involved.

**Weaknesses:**

The benchmarks are very specific to visual search and quite a lot of effort has been put into assessing a disparate range of models on the same criteria. It would be interesting to comment on how these benchmarks might be applied beyond the immediate domain, and how the lessons learned on unifying analysis could be applied when working on related problems.

**Additional Feedback:**

Minor points and queries:

Table 1: Mean scanpath length in the unrestricted case is presented as 10.7 +/- 10.9 which would give a lower bound of less than zero. Is this a typo?

Table 1: Where numbers are presented with bounds, are these bounds the absolute range, or something else (eg. 1 s.d.)? It may be worth clarifying this.

Page 5, Line 125: 'in order to engulf them in a common framework': 'engulf' was a surprising turn of phrase. Perhaps 'place' would be better.

It may be worth highlighting on first reference that references to tables / figures prefixed 'S' are in the supplementary material section.



**Clarity:**

The paper is generally very well-written and easy to understand.

One minor criticism is that there are a relatively large number of domain-specific acronyms which are used in a way that might hinder understanding in some cases (eg. hmMM and whMM are defined but the exact acronyms are never expanded, SSIM used without being defined).

**Correctness:**

I am unable to make any strong claims about the correctness of the analysis as I don't have enough familiarity with how common metrics are computed in other visual search tasks.

**Documentation:**

Code is shared in a github repository, under the MIT license, and includes the datasets and implementations of the models used. Where datasets are under different licences, this is generally made clear in the repository with a link to the licence. In the case of the CVLab @ Stony Brook datasets the licence is not noted explicitly in the GitHub repository and appears to be more restrictive than the MIT licence. The licences for the third-party data are all stated in section C of the supplementary material, but the exact details of the licence under which the pre-processed data from Stony Brook are being re-shared should be clarified in the repository.

I tested that running the code in the repository (on branch `main` at commit `7503f63`) yields the same numbers as in Table 2, leading me to believe that the code allows complete reproduction of the results in the paper. Running the code was easy, following the instructions in the README file worked flawlessly (modulo having to downgrade the pip package for numpy by one minor version to avoid a version conflict error with another package).

In the README in the github repository, the exact configuration of the code used for the SVRHM workshop submission at NeurIPS 2021 is noted as being in a specific branch of the repository. The authors should consider doing the same, either with a git branch or tag, to allow reconstruction of the results in this submission as the codebase evolves.



**Ethics:**

The datasets and models analysed in the paper include data collected from human subjects, and pre-processed datasets are made available via the GitHub repository. However, this submission only re-analyses existing published data and does not contribute any new data from human subjects. As noted above, the paper and repository references all the original datasets and model, and links are provided to where they are available and their licences.

**Relation To Prior Work:**

The authors provide clear references to the datasets and models which are analysed in the paper. The paper's main contribution is to highlight the challenges and opportunities presented by unifying the analysis of various pieces of prior work.

**Summary And Contributions:**

The authors propose a standardised procedure for evaluating both human-derived datatsets and models of visual search in a single framework. The procedure is evaluated on a set of publicly-available models and datasets and the results are discussed. In providing a unifying framework for evaluating models and datasets, the authors aim to make it possible to test disparate models and datasets within a standard set of criteria. The paper builds on a poster presentation from the SVRHM workshop at NeurIPS 2021. The exposition is expanded significantly, and a great deal of supplementary material (which I did not review in detail) is provided.

Overall, I believe this paper constitutes a valuable move to provide a standard set of analysis criteria for visual search and a standardised framework in which to perform this analysis, highlighting some of the challenges faced by a suite of existing models.

For clarity, I am not a domain expert in visual search, and so I am unable to make strong statements as to whether the analysis is complete or whether there are missing references to other common datasets and benchmarks - hence my low confidence score. My review is based on my understanding of the technical merits of the analysis, the overall clarity of the exposition and the quality of the accompanying code and documentation.

---

### Official Review · Reviewer_sZ1v · 2022-07-27
**Paper highlights critical issue in visual search research, but paper structure and focus detracts from contributions**

**Rating:** 7
**Confidence:** 3

**Strengths:**

- The motivation is compelling and clear.
- Experiments are well thought out and clearly described.
- Includes a thorough analysis of model differences.
- Appendix is extensive.
- Introduction describes background and necessary knowledge that motivates the primary contributions of the paper.

**Weaknesses:**

- The paper does not fully explain why the authors made many of the choices described in the paper, such as model/dataset selection, metric selection, etcetera. This may cause readers to question the efficacy of the proposed evaluation framework.
- Given that the evaluation framework is a key contribution of the paper, it would have been nice to include a more extensive discussion detailing the benefits and downsides of using the chosen metrics.
- Given that producing a framework for model comparison is a primary contribution, it would be helpful to possibly include a full section detailing how the models are modified to allow direct comparison.
- See "clarity" section.

**Additional Feedback:**

The work described in the paper seems well done and useful to the community. Revising the paper to more clearly illustrate the contributions and explain the authors’ thought processes would highlight the very positive aspects of the work.

**Clarity:**

- Overall, the paper could benefit from more structure. As the paper is, important points are buried in paragraphs of details. This is especially true in the "results" section.
- It would have been helpful for section 2.1 to begin with an explanation/review of what datasets were chosen and why (e.g. it is unclear why MCS was included, especially since it does not fit the restrictions introduced in the conclusion) before delving into details – as it is, this section is possibly disorganized and hard to follow. The important differences between the datasets get lost in the details, and Table 1 doesn't provide particularly useful information to assist with this.
- A quick overview of the metric abbreviations in the caption of Table 2 would be helpful for legibility.
- The restrictions mentioned in the second paragraph of the conclusion would have been helpful if mentioned earlier in the paper.
- It is not explained why the baseline models in 2.2.2 were chosen.

**Correctness:**

Yes, overall the claims made in the paper are correct and backed up by provided data. The experiments are well thought out and run, and the appendix is very comprehensive and detailed.

**Documentation:**

The provided repository is complete and has excellent documentation.

**Relation To Prior Work:**

While much related work is touched on in the introduction, it might be helpful to have a section that gives an overview of the visual search models that exist to provide context for why the three models were chosen, and a separate overview of existing visual search datasets. As the paper is, it is difficult to understand why the models and datasets were chosen.

**Summary And Contributions:**

The authors produce a framework and metrics for comparing three visual search models and use these to identify differences between the approaches. They adapt the Ideal Bayesian Searcher model to allow it to work with a wider range of datasets, and highlight how using a unified framework can produce better algorithms in general.

---

### Meta-Review · Area_Chair_BQA5 · 2022-09-04

**Recommendation:** Accept
**Confidence:** 4

**Metareview:**

There are extensive discussions between authors and reviewers. The authors have done an excellent job addressing the raised issues during the rebuttal phase. As such, reviewers raise the scores. Overall, there is sufficient support to accept this paper from the reviewers. The authors should include the replies to reviewers and revise this paper before the conference date.

---

### Decision · Program_Chairs · 2022-09-16

Accept